# Global risk of wildfire across timber production systems

Christopher G. Bousfield [1,2] ✉, Oscar Morton [1,2], David B. Lindenmayer [3], Adam F. A. Pellegrini [1,2], Matthew G. Hethcoat[4] & David P. Edwards [1,2]

Timber is worth $1.5 trillion US Dollars annually with demand rising, but wildfires increasingly threaten production. Plantations occupy 3% of forests globally and produce 33% of the world's timber, but a critical question is whether they are more vulnerable to stand-replacing wildfires than natural production forests. We combine forest management and wildfire data to estimate that 15.7 (14.7–16.7) million hectares of natural production forests and 1.4 (1.26–1.64) million hectares of plantations suffered stand-replacing wildfires between 2015 and 2022. Using statistical matching for 17 countries representing 50% of global production and 75% of burned timber-producing forest, we find plantations in temperate regions were twice as likely to suffer stand-replacing wildfires than natural production forests, including in vital timber-producing nations like China and Russia. Plantations in tropical regions showed no clear effect, with national differences ranging from 75% lower to 58% higher risk of burning. Given increasing global reliance on plantation timber, preventing wildfires through landscape-level planning, fire management, and increased plantation diversity is critical for global wood security.

Timber is a critical natural resource to the global economy, with production worth ~$1.5 trillion US Dollars per year[1]. One third of the world's forests are used to produce timber and the forestry sector employs over 33 million people worldwide[2]. Demand for timber is expected to increase by 54–200% by 2050[3,4], due to increased urbanisation and population growth, and global shifts away from carbon-intensive construction materials, such as steel and concrete, as countries move towards net-zero economies[5].

Climate change is expected to cause increasingly frequent and severe weather extremes, which will threaten forests and their ability to provide timber and other ecosystem services[6,7]. Between 2001 and 2021, wildfires caused the loss of 19–25 million hectares (Mha) of timber-producing forests globally, with annual wildfire-induced losses increasing in many important timber-producing regions[8]. A key challenge is to identify ways to ensure continued supplies of timber despite the increased threat of wildfires to timber-producing forests. The future production of timber,

however, depends not only on the amount of forest that burns, but the type of forest that is burning.

There are two primary methods of producing timber: through the harvesting of naturally occurring native forests that produce wood and regrow naturally after logging, or through intensively managed timber plantations composed of one or two species of even ages with regular spacing[2]. Plantations currently occupy ~3% of forests globally[2], yet supply over a third of global timber[9], expanding in area by 75% since 1990[2]. Plantations also produce timber at a much faster rate than natural forests, with wood volumes on average 220% higher than native forests of the same age[10,11]. Given their rapid expansion in area and fast growth rates, plantations represent an important opportunity to meet future timber demand whilst protecting remaining native forests from degradation through timber extraction[12–14].

A crucial risk of plantations is the potential that they are more prone to increasingly frequent and severe wildfires. Plantations are low diversity, often containing one or two species initially planted in

[1]Department of Plant Sciences and Centre for Global Wood Security, University of Cambridge, Cambridge, UK. [2]Conservation Research Institute, University of Cambridge, Cambridge, UK. [3]Fenner School of Environment and Society, The Australian National University, Canberra, Australian Capital Territory, Australia. [4]Canadian Forest Service, Edmonton, AB, Canada. ✉e-mail: cgb48@cam.ac.uk

densely packed rows[15]. This uniform stand structure may make some plantations more susceptible to extreme environmental conditions than natural production forests that contain a mix of species and age groups[16]. Dense plantations of non-native trees such as *Eucalyptus* and *Pinus spp.* have been implicated in large recent forest wildfires in Portugal and Chile[17]. However, global-scale evidence of the link between wildfires and timber plantations is lacking, indicating an urgent need to understand the relative fire susceptibility of plantations and natural production forests in a future with more wildfire.

Here, we combine global datasets detailing the spatial extent of the two key forms of timber production—natural production forests and timber plantations (see Table 1 for definitions)[18]—and stand-replacing wildfires between 2015 and 2022[19]. Given the location of natural production forests and timber plantations varies globally and regionally and span diverse environments, we use statistical matching and modelling to control for environmental and anthropogenic covariates that could influence fire occurrence. We do so to address two questions: First, how have recent wildfires impacted both natural production forests and timber plantations? Second, under identical fire-influencing environmental and anthropogenic conditions, are plantations more likely to burn in wildfires than natural production forests? We find widespread burning of both natural production forests and timber plantations, and elevated burn risk in plantations across temperate regions.

## Results

### Spatial distribution of wildfire-driven losses of natural production forest and plantations

Globally, natural production forests extend over 2.1 billion hectares, compared to plantations which cover 289 Mha[18]. Natural forests used for timber production are widespread across temperate, tropical, and boreal regions (Fig. 1a), but plantations are concentrated primarily in the USA, Europe, China, and Brazil (Fig. 1b). Between 2015 and 2022, 15.7 (s.e.m. 14.7–16.7) million hectares (Mha) of natural production forest and 1.40 (1.26–1.64) Mha of timber plantations were lost to stand-replacing wildfires.

The vulnerability of natural production forests and plantations to stand-replacing wildfire varied substantially among countries and regions (Fig. 1c, d). Natural production forests experienced the highest burn rates (Supplementary Fig. 1) in Brazil (3.69 Mha, 3.47–3.95 Mha), the USA (3.08 Mha, 2.92–3.19 Mha; mostly in the west), Australia (2.85 Mha, 2.80–2.94 Mha), Canada (1.41 Mha, 1.35–1.46 Mha), and Russia (1.18 Mha, 1.08–1.28 Mha). Plantation burning was highest in the USA (0.20 Mha, 0.16–0.25 Mha), Australia (0.16 Mha, 0.15–0.17 Mha), Canada (0.13 Mha, 0.13–0.13 Mha), Portugal (0.13 Mha, 0.12–0.14 Mha), and Chile (0.12 Mha, 0.12–0.13 Mha). Large areas of both natural production forest and plantations remained largely unimpacted by stand-replacing wildfires across south-eastern USA, Northern Europe, and China (Fig. 1c, d). The divergent responses across the various regions raises a need to consider variables that influence wildfire risk.

### Elevated burn risk in plantations under identical conditions

To quantify the difference in burn probability between natural production forest and plantations, we used statistical matching to account for the impact of confounding variables that might influence fire likelihood in timber-producing forests, including topography, weather, and proximity to anthropogenic activities. After applying rigorous matching methods (see Methods, Supplementary Fig. 3) within each of the top 50 timber-producing countries[20], we retained 11 temperate countries and six tropical countries containing both plantation and natural production forest. These nations included the four largest global timber producers (USA, China, Brazil, and Russia), and together account for 50% of global timber production[20] and 75% of global burned timber-producing forest (12.7 M ha, Supplementary Fig. 1) in the study period (2015–2022).

After matching and modelling to isolate only the effect of management type (natural production or plantation) on burn risk, we found differing effects of temperate and tropical plantations on fire probability (Fig. 2a). Temperate plantations were on average twice as likely to burn as natural production forests under the same conditions (100% higher burn probability, Fig. 2b red line; $p < 0.001$). At the national scale, this effect was observed in 9 of the 10 temperate countries, spanning Europe, temperate Asia, and Australasia, with national averages ranging from 57 to 155% higher burn probability in plantations (Fig. 2). Higher likelihoods of plantations burning were present in globally important timber-producing nations, including China (122%) and Russia (117%), the second and fourth largest timber producers, respectively[20]. The same patterns were present in most of the temperate countries assessed, including France, Italy, Spain, and New Zealand (122–130% increases).

Whilst the absolute difference in burn probability was relatively small in most countries (<1%, Fig. 2a), the scale of timber production in these nations (e.g., >800 Mha in Russia, >200 Mha in China) means that even a small shift in burn probability can result in large areas of additional burning. The greatest difference in absolute burn probability was found in Portugal where plantations had a burn probability of 16.8% (CI95: 15.0–18.7%) compared to 8.2% (CI95: 7.0–9.5%) in natural production forests between 2015 and 2022, supporting suggestions that widespread *Eucalyptus* plantations were responsible for recent unprecedented wildfire events[17]. Higher plantation burn risk is of particular concern for timber production in Mediterranean countries where some of the most pronounced increases in extreme fire weather are expected to occur in the future[21,22].

The USA was the only temperate country to show no significant difference in burn probability between plantations and natural production forests during the period, possibly because most plantation points are in south-eastern USA where low-intensity and frequent prescribed burns are common[23], and stand-replacing wildfires are less likely[24]. Matched natural production points are likely biased towards forestry operations in the east where wildfire is less prevalent than in the north-west[24], with regional studies in the USA finding a greater likelihood of high-severity burning in intensively managed forests[25,26].

Tropical plantations showed no overall effect on burn probability ($p = 0.87$), but this conceals considerable national-level variation. Australia, South Africa, and Indonesia demonstrated a significantly lower wildfire probability in plantations compared to natural production forest (7–75% less likelihood). This reduction was most pronounced in Australia, where matched natural production forests were four times more likely to burn than plantations, supporting research suggesting that timber harvests increase fire risk due to the creation of ground-level fuel and transitions to younger, less fire-resistant forest stands[27–29]. Recent bans on native forest logging in the states of Western Australia and Victoria will likely contribute to protecting future timber supplies under increasing wildfire risk.

Of the remaining tropical countries, Brazil, Argentina, and Vietnam demonstrated no effect of forest production type on the likelihood of stand-replacing wildfires, whilst Uruguay had increased likelihood in plantations versus production forest. In Brazil, burn rates are relatively high for both production types, with the high likelihood of wildfire in *Eucalyptus* plantations across southern and central Brazil similar to high burn rates in natural forestry systems on the edges of the Amazon after recent record wildfire seasons[30]. In addition, mapping of selective logging systems using remote sensing in tropical regions such as Brazil and Indonesia remains a challenge[31], making it harder to accurately deduce differences between forest management classes.

Assessing wildfire risk at country scale is important since wood production practices, policies, and management differ between nations, as do volumes of timber production and the balance between imports and exports. Key countries at higher risk of wildfire-driven

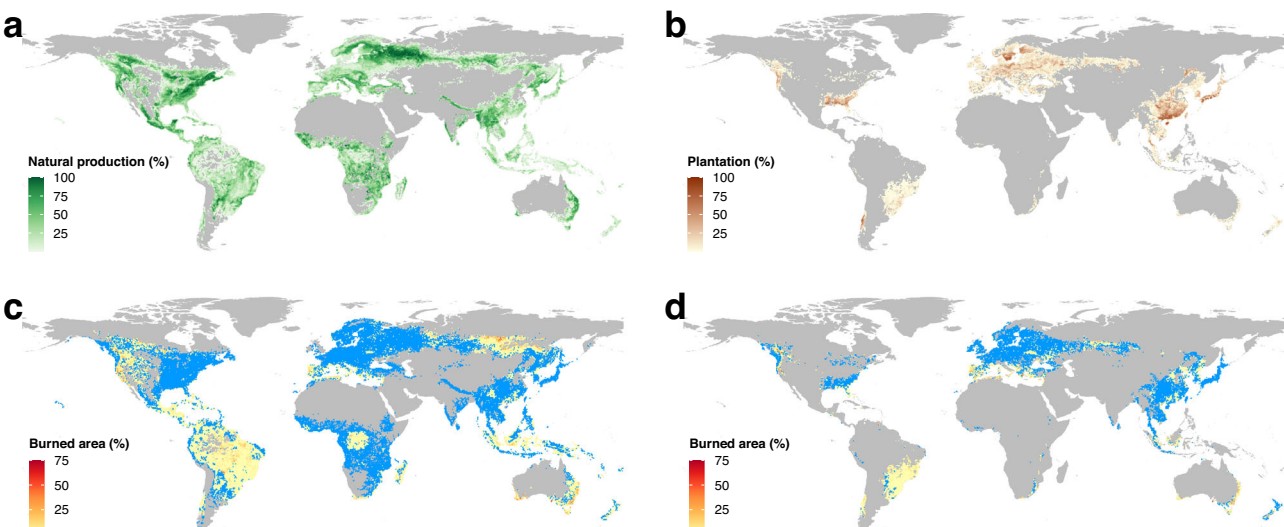

**Fig. 1 | Global distribution of forestry land and wildfire-driven forest loss.** Total proportion of each one-degree grid cell mapped as natural production forest (**a**) and timber plantation (**b**) by Lesiv et al.[18], and total proportion of natural production forest (**c**) and timber plantation (**d**) in each cell lost due to wildfires between 2015 and 2022[19]. Areas mapped in grey have no or limited mapped logging activity. Areas of warmer red indicate greater burned area, areas mapped in blue have no or limited burned area (<0.1%) in timber-producing forest.

production losses and thus wood insecurity can be identified by considering their relative reliance on plantations for timber alongside the change in stand-replacing wildfire risk compared to natural production forests (Fig. 3, see Supplementary Fig. 2 for version with FAO data and Supplementary Table 1 for comparison). China has particularly high risk given its high relative burn effect of plantations (122% more likely than natural production forest) and large reliance on plantation production (45% of all timber-producing forest, Lesiv et al.[18]), further exacerbated by the 2017 ban on natural forest logging. While this ban has benefits for biodiversity and climate through sparing native forest from degradation[13] and expanding forest cover[32], it elevates the fire-related insecurity in China's timber-production chain. Other countries less vital to global timber production have similar risk profiles given their reliance on plantations, including Ukraine, Belarus, and Portugal, whilst restrictions on natural forest harvesting means that >99% of timber in New Zealand is now sourced from plantations[33]. Conversely, Russia has a high plantation burn effect but low reliance on plantations for timber, reducing the fire risk to national timber security. Production in Australia, South Africa, and Indonesia is also at risk since natural production forests are significantly more likely to burn than plantations, yet they accounted for 94–95% of their timber-producing forests by area (this will drop substantially in Australia in future years given the shift from natural production).

The map of forest management[18] allows for global-scale comparisons of fire risk in natural production forests and plantations, but mapping timber harvests globally at such fine resolution is difficult, and these data are not without uncertainty. Whilst the overall accuracy of the map is 83%, the user's accuracy for the forest management classes used in this study range from 58–71%, with the extent of temperate plantations often underestimated and confused with natural production forests. To counter this, we included only mapped pixels with a classification confidence ≥70%, which represented a compromise between high-confidence classifications and ensuring enough data points to allow for rigorous statistical matching and modelling. Mapping of different timber production systems is also difficult due to ambiguous definition of production types within the FAO (Food and Agriculture Organization of the United Nations) framework, particularly in the case of 'planted forests' and 'plantation forests' where the specific differences between the two, and whether both typically produce timber, is not always clear. The FAO definition of forest consisting of only 10% tree cover also raises difficulties, since this definition will include many savannah areas[34] with limited timber value. Moving towards improved and more specific definitions of forest management, alongside globally consistent annual satellite-based mapping of timber harvests, would allow for dramatic improvements in our understanding of timber harvest dynamics and future threats to global timber production.

## Discussion

Stand-replacing wildfires have driven widespread global losses of timber (17.1 Mha) in both natural production forests (15.7 Mha) and timber plantations (1.4 Mha) between 2015 and 2022. We found that temperate plantations were twice as likely to burn than natural production forest under the same conditions, whilst tropical plantations showed no overall effect on burn likelihood despite high national-level variation. Plantations play a vital role in meeting global timber demand, but given that forest wildfires have been increasing globally[8,19] and future climate change is predicted to increase fire extent and severity[22], the risk of wildfires in plantations will only continue to grow. Improving fire management in timber plantations will be crucial in meeting global timber requirements and avoiding substantial timber losses.

Why temperate plantations are more likely to suffer stand-replacing wildfires than natural production forests varies globally, but the dense, homogeneous, and highly connected forest structure of plantations likely plays a role[35]. Shorter rotation times in plantations than natural production forests mean they are often managed as regrowth stands, with cohorts of young to intermediate-aged trees that are more susceptible to high-severity burning (for mechanisms, see ref. 36) and unable to grow to a more mature and fire-resistant state[37]. By contrast, native forests are usually older, contain a range of tree sizes, species, and spatial structures that can make them less susceptible to high-severity burning[25,36]. In addition, low-diversity understories in exotic timber plantations[15,38] can harbour invasive species[39], which increase fire risk[40]. The dominant tree species used also plays a role in fire proneness. In 2017, record-breaking wildfires in Chile burned greater proportions of *Pinus* plantations than *Eucalyptus*[41], whilst in the record 2019/20 Australian wildfire season, conifer plantations suffered higher incidences of fire-induced severe canopy damage compared to *Eucalyptus* plantations, potentially due

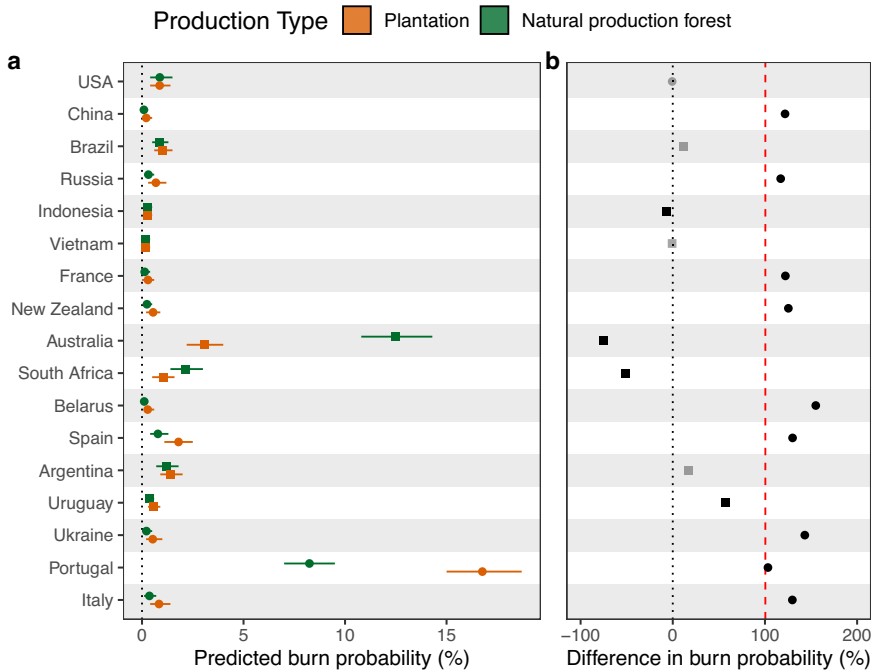

**Fig. 2 | Modelled burn probabilities of natural production and plantation points after matching for fire-influencing covariates.** Predicted burn probabilities for natural production forest and tree plantations between 2015 and 2022 (**a**) and the percentage difference between the predicted mean burn probabilities of the two timber production types (**b**). Natural production forests are represented in green, whilst plantations are represented in brown. Countries are ordered by their volume of timber produced, with temperate countries represented with a circle whilst tropical countries are represented with a square. Black points in (**b**) represent countries with significant differences in burn probability between timber production types, grey points represent countries with no significant difference. Red line in (**b**) represents the mean difference in burn probability between plantations and natural production forests across all 10 temperate countries ( +100%, $p < 0.001$); no line is present for plantations in tropical countries since no overall effect was detected. Error bars are 95% confidence intervals from 10,000 simulations of 1000 randomly selected forest points.

to the higher live fuel load and density in *Pinus* tree crowns[42]. Understanding potential variance in fire proneness between different plantation species is therefore an important next step in minimising fire risk across global plantation systems.

Despite their increased susceptibility to wildfires, timber plantations produce a harvestable crop more rapidly than natural production forests, creating a trade-off between fire risk and productivity. For example, in Spain, plantations are ~2.5 times more likely to burn than natural production forest, but can achieve 3–4 rotations in the same time period as one natural production forest rotation (35 years rotation in exotic pine plantations versus 110–145 years in native broadleaved oak[43]). In Australia this trade-off is even clearer, given plantations have a significantly lower burn risk and produce sawlogs in 20–30 years compared to 80 years for natural production forests[44]. Plantations may also represent a more reliable strategy for production in countries where fire frequency is generally high across production types (e.g., USA, Brazil, Portugal) and will increase further with climate change. Understanding the relationship between fire risk and rotation time in different regional contexts is of high importance in planning future timber-production strategies.

Timber plantations are rapidly expanding and already contribute >33% of global roundwood supplies[9]. Our results demonstrate that increased reliance on plantation-based harvests will require further interventions to combat their elevated fire risk. At the global scale, plantation expansion should be concentrated in areas where productivity is high, and absolute fire risk is relatively low[37], provided plantation expansion does not replace natural ecosystems[10], and could instead occur on low-intensity or abandoned agricultural land[45]. At the landscape scale, plantations should be planted in flatter areas, designed in mosaics of mixed species and ages, interspersed with other less flammable land-use types (e.g. riparian vegetation, grazing

lands) and green firebreaks[37], and located away from population centres to minimise human exposure risk[46].

The variability in national-level impacts suggests that fire management in plantations may be effective at combatting any increased burn risk in some countries (e.g. USA, Australia). Extension of these fire-management practices (e.g., prescribed burning, thinning[23,37]) and roll-out of emerging technology (e.g., drones and camera arrays, autonomous watergliders, etc.[37]) should be considered, within national-level contexts and fire regimes. Promoting diversity within plantations will also become increasingly important under climate change[6], with mixed-species plantations more resilient to disturbance[16]. Despite the benefits, including increased yields and biodiversity[11], low-diversity plantations (1–2 species) still dominate current planting and future tree-planting commitments[47], perhaps due to lower management costs.

In conclusion, our results demonstrate the elevated risk of wildfire-induced timber losses under plantation systems compared to natural production forests, which will become increasingly important as climate change leads to higher frequency and severity of wildfires. However, plantations are a key and rapidly expanding element of the global timber system[2], representing the most efficient means of producing timber[10]. Minimising the risk of fires in plantations through effective regional and landscape-scale planning, fire management practices, and shifting towards more diverse plantations[16] will be critical in maintaining global wood security in an increasingly hostile climate.

## Methods
### Data sources
For global data on forest management, we used the map produced by Lesiv et al.[18] which contains data on forest management type across the

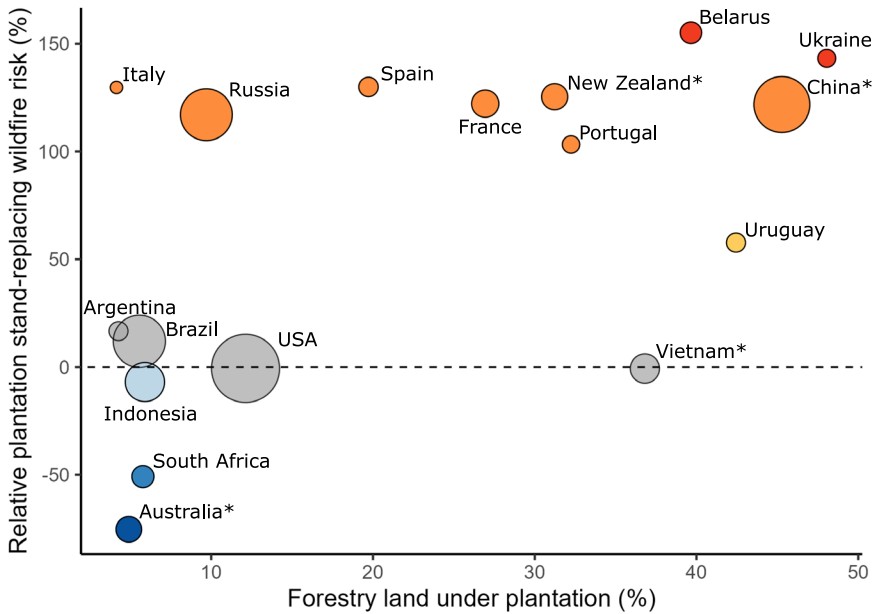

**Fig. 3 | National-level plantation wildfire risk effect size and percentage of timber-producing forest under plantation systems.** Plantation stand-replacing wildfire risk represents the relative mean percentage change in wildfire probability in plantations compared to natural production forests under the same wildfire conditions. Points in orange to red represent increasingly high plantation wildfire effect (i.e. plantations are increasingly more likely to suffer stand-replacing wildfire than natural production forest), points in darker blue represent increasingly low plantation burn effect (i.e. plantations are increasingly less likely to suffer stand-replacing wildfire than natural production forest), and points in grey demonstrate countries with no significant plantation burn effect. Percentage of timber-producing forest under plantation systems is from Lesiv et al.[18]. Points are sized by their share of global timber production between 2015-2022 according to the FAO[20]. Countries with * are those that have banned native forest logging either at the national (China, Vietnam, New Zealand) or state level (Australia) and now source the majority of their timber from plantations.

**Table 1 | Definitions of timber production types used in this study and Lesiv et al**

| Term – This study | Class in Lesiv et al. | Definition |
|---|---|---|
| Natural production forest | Naturally regenerating forest with signs of management e.g. logging, clear cuts etc. | Forest is managed with signs of logging (including selected logging) in the 100 m pixel or nearby, but there are no signs of planting. |
| Temperate plantation | Planted forest | Forest is managed and there are signs that the forest has been planted in the 100 m pixel. Rotation time is relatively long ( >15 years). |
| Tropical plantation | Plantation forest | Intensively managed forest plantations for timber with short rotation (15 years max). |

world's forests at 100 m resolution for the year 2015. The map was produced using PROBA-V satellite images and trained classification algorithms to group the world's forests into seven different management types, of which three are included in this study. The first is "Naturally regenerating forest with signs of management, e.g., logging, clear cuts etc", which we use to represent our natural production forest areas. For plantations, we use two other categories from Lesiv et al. The first is "Plantation forests (rotation time up to 15 years)", which are short-rotation plantations employed solely in tropical countries, and we refer to these as 'tropical plantations', and the second is "Planted forests", which are longer-rotation plantations located in temperate countries, which we refer to as 'temperate plantations'. The difference between these two categories is largely associated with location and rotation time (<15 years for "Plantation forests", >15 years for "Planted forest", see Table 1 and Lesiv et al.[18] for original definitions), and we use both types to represent timber-producing plantations globally.

For the fire data, we were interested in high-severity wildfires that caused significant damage and a loss of forest cover that can be assumed to also cause a loss of marketable timber. For this part of our analysis, we used the map of forest loss due to fire by Tyukavina et al.[19], which extends the work of Curtis et al.[48] and uses MODIS and Landsat imagery to identify forest loss events (defined as the loss of woody vegetation >5 m high) that were directly caused by wildfires in the period 2001–2022. Following Bousfield et al.[8], we chose not to use the

MODIS Burned Area dataset directly since these wildfires do not differentiate between stand-replacing fires and non stand-replacing fires. In many regions low-severity burns such as those mapped by MODIS may not cause significant losses of timber where they occur, or may even be actively employed in forest management as part of prescribed burning regimes[23].

## The impact of recent wildfires on production forests and plantations

To understand where natural production forests and tree plantations have burned in recent years, we intersected spatial data on forest-loss causing wildfires[19] with global data on forest management type[18]. We overlayed the map of forest-loss due to fire with the map of forest management and calculated the total spatial extent and distribution of stand-replacing fires in timber-producing forests, splitting the timber-producing forests into two broader categories, natural production forests, and timber plantations (either tropical or temperate). Since the forest management map is centred around the year 2015, we included only those fires that occurred post 2015 in our analysis. Spatial overlays were conducted using the R packages *sf*[49] and *terra*[50].

## Relative burn risk in plantations and natural production forests
We used the same three classes of forest management from Lesiv et al.[18] as outlined above. Given the levels of uncertainty in pixel classification

between management types and geographical locations in Lesiv et al.[18], we first used the confidence matrix in Lesiv et al. to subset only the pixels with a classification confidence higher than 70%, to ensure our data points were largely accurate whilst retaining enough data for a global analysis with good spread across countries. We focussed on the 50 largest timber producing countries according to the FAO[20] as our study regions, and filtered the Lesiv et al.[18] data to include only those pixels within these country boundaries. Given the size of some of these countries (e.g., the USA, Russia) and the large number of pixels, we opted to take subsamples of the timber-production pixels to allow for analysis in a tractable timeframe. To ensure good sampling across the whole range of each country, and to improve the quality of matching (which was poorer under standard random sampling), we divided each country into one-degree grid cells, and randomly sampled without replacement up to a maximum of 5000 pixels of each forest management type in each one-degree cell. To remove the influence of low tree cover systems that burn regularly but do not produce significant amounts of timber (i.e., savannahs), we then filtered the samples for those that contained ≥30% tree cover (following the definition of forest in previous research[51–53]). We then pooled the samples from all grid cells in each country together to form our country-level sample of forest management points.

## Matching

We undertook statistical matching as a pre-processing technique to reduce bias in covariates that could affect the probability of fire or timber management type, considering a total of ten environmental and anthropogenic variables (see Supplementary Table 2. for source details). Using the *MatchIt* package[54], we employed nearest neighbour matching for the following covariates: elevation[55], slope[55], mean temperature of the hottest quarter, mean precipitation of the driest quarter and mean precipitation of the wettest quarter from the Worldclim dataset[56], the 95th percentile of the Fire Weather Index based on ERA-5[57], burn area history in the previous 15 years[58], landscape level tree cover[51], distance to the nearest road[59] and population density[60]. We used exact matching for biome[61].

We conducted the matching on a country by country basis, and attempted to match natural production forest points with plantation points in each of the top 50 timber-producing countries. We deemed a mean covariate balance of <0.25 for each covariate as acceptable, and dropped any country whose matched dataset did not achieve this score for all covariates. Following the guidelines of Schleicher et al.[62] and methods used in recent studies[63,64], we tested two different common matching algorithms (Mahalanobis and propensity score) with two different callipers (0.2, 0.5), and selected the most effective matching method based on the results of diagnostic checks for covariate balance, which was propensity score matching with a calliper of 0.2 (Supplementary Fig. 3). This method left us with matched datasets of natural production and plantation forest management types from 30 different countries for further analysis. See supplementary information for exact matching code used.

## Modelling

Whilst matching seeks to remove covariate imbalance between treatment groups, it cannot completely eliminate such imbalances across the dataset. We therefore undertook further modelling to resolve this issue. We fitted GAMMs (Generalised additive mixed models) using the *mgcv* package[65] to our matched datasets, with forest management type as a parametric term, our numeric covariates used in matching as cubic regression smoothing splines, and additional X and Y coordinate splines to account for spatial autocorrelation. We also included random intercepts for both country and biome, and interaction terms between biome and country, biome and management type, and biome and tree cover. Given the size of our global dataset, we used the bam function with the default fREML method, and the binomial family with

logit link (see supplementary information for exact code used). All numerical covariates were centred and standardised prior to model fitting to aid stable convergence. To constrain our analysis to countries where wildfires are a significant problem, and improve model performance, we included only countries where the burned area represented >0.1% of matched timber-producing forest points. This left 17 countries for the final modelling step. Given the inherent difference in plantation systems, natural forest equivalents and fire regimes between plantations in temperate countries (termed "planted forest" by Lesiv et al.) and tropical countries (termed "plantation forest" by Lesiv et al.), we fit two separate models, one featuring all temperate plantations and their matched natural production forest points, and the other featuring all tropical plantations and their matched natural production forest points. We assessed spatial autocorrelation post-fitting by examining autocorrelation plots of model residuals, finding no evidence of spatial autocorrelation in all countries except the four largest (USA, Russia, China and Brazil), where some autocorrelation was observed likely due to their sheer size and diversity of sub-national practices.

To isolate the effect of forest management type, we used a counterfactual approach, whereby we compared the simulated outcome (burning or non-burning) of each point under both natural production and plantation management. This approach specifically identifies the difference between the two practices under the points' true conditions (e.g. elevation, temperature etc.). To do this we randomly sampled 1000 forest points per country, with their associated covariates and coordinates, assigned each sampled point to both a natural production and plantation treatment, and simulated the outcome (burn or non-burn) under both treatments. We then summed the predicted burned area (of the 1000 points) under both the natural production and plantation treatment and calculated the difference in burned area between the treatments. We repeated this simulation for 10,000 samples of 1000 points per country and summarised this distribution of treatment differences to a mean difference in predicted burn probability (and 95% confidence intervals). The data presented in Fig. 3 thus represent the mean (and 95% confidence intervals) predicted burn probability for matched natural production and plantation forest points under identical conditions.

## Data availability

All data used in the study are publicly available online. Tyukavina et al.[19] fire data (https://glad.umd.edu/dataset/Fire_GFL/), Lesiv et al. forest management data (https://zenodo.org/record/5879022#.ZCa9IXbMKUk)[18]. Data used in the matching analysis can be found in the Supplementary Information.

## Code availability

Code for the analysis can be found in the supplementary information.

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

## Acknowledgements

Funding was provided to D.P.E. from the Natural Environment Research Council (grant no. NE/W003708/2).

## Author contributions

Conceptualization: C.G.B, D.P.E., D.B.L., M.G.H and A.F.A.P. Methods: C.G.B and O.M. Investigation: C.G.B. Visualization: C.G.B. Supervision: D.P.E., D.B.L, A.F.A.P. Writing – original draft: C.G.B. Writing – review and editing: all co-authors.

## Competing interests

The authors declare no competing interests.
