## [Transparent Peer Review file · Nature Communications]

Global risk of wildfire across timber production systems

Corresponding Author: Dr Christopher Bousfield

Version 0:

Reviewer comments:

Reviewer #1

(Remarks to the Author)

This paper addresses an increasingly common question about fire proneness of various types of forests and it's very good to see an attempt to compare natural forest systems with plantations. The use of the Lesiv approach to access data on the different management types of forests gives a good opportunity to address this question globally and the matching approach allows proper comparisons under similar conditions.

The paper is important as it is the first attempt at comprehensively answering the question as to whether plantations have a higher fire incidence than natural production forests under the same conditions– but it requires some caveats.

I query the data for Belarus and Ukraine and their reported proportion of plantations in Figure 3. Looking at FRA data for 2015 Belarus has only reported 6040 hectares of plantation and a total forest area of 8.633 m hectares – or 0.1% plantations, Ukraine has 368,000 ha of plantations with 9.65 m hectares of total forest or 3.8%. Some variation is to be expected as resource data sources are different (FRA and Lesiv) – but this is very large.

New Zealand is also mis represented – plantation timber production makes up nearly 99% of national production – natural forests in public ownership are managed purely for conservation purposes and only a very small area of privately owned natural forest is managed for timber. This suggests misclassification in the Lesiv paper.

It would be helpful and give readers context if there was a summary table of area of naturally regenerating production forest, tropical plantations and temperate plantation (the three management types used) for the countries in a supplementary table. I suggest review of Figure 3 and working out how best to make the important point about the observed increasing risk with increasing dependence on plantation sourced timber.

The paper should include some comment in the discussion about uncertainty in the findings – the Levis paper notes it is ‘the first reference data set and a prototype of a globally consistent forest management map’ and ‘with forest management class accuracies ranging from 58% to 80%’ – and this uncertainty will carry through to the analysis in this paper. See my comments on differences between this data and FRA data above.

I found Figure 1 to be too small to properly view the images and suggest it be split into two.

Be consistent – natural or native production forests

Line 85. Potentially modify this to say plantations are low diversity often containing one or two species initially planted in densely packed rows as species source may not be all that relevant in the description it's not so much native versus exotic as species fire susceptibility.

Reviewer #2

(Remarks to the Author)

Bousfield et al. examined wildfire risks across timber production systems in the major timber production counties in the

world. The forests were divided into the natural production forest and plantation forest. GAMMs was fitted and implemented to estimate the burn probability. Forest management data was from Lesiv et al., which contains data on forest management type across the world forest at 100 m resolution. The method is sound. My major concern is how this paper is different to their previous work published in 2023.

Bousfield, Christopher G., David B. Lindenmayer, and David P. Edwards. "Substantial and increasing global losses of timber-producing forest due to wildfires." *Nature Geoscience* 16, no. 12 (2023): 1145-1150.

In Figure 1, it is not clear what the blue color in C and D denote.

Reviewer #3

(Remarks to the Author)

Article "Global risk of wildfire across timber production systems" deals with a very interesting topic. The questions raised in the article are indeed worth of discussion worldwide. The article fits well in the scope of the journal and is indeed worth publishing. I feel this will increase people's understanding of the role of fires in forestry and in relation with global wood markets. The paper is generally well-written, used analyses sound, and the given results rely on the literature cited well.

However, I include number of suggestions that might improve the paper.

I suggest to include few lines & include critical discussion about FAO:s forest classification (that do have many inaccuracies) that cause some possibilities also to have errors when interpreting global forest data. FAO:s forest classification is based on too loose definitions.

Terminology used in this ms is somewhat confusing, and for exact definitions, you need to look for them in the methods-section. Ms could be improved by being more exact with the terminology. A simple table with terminology & definitions used might help the reader. Or at least short definition in the main text. Suggest using the defined terminology throughout the paper. An example, lines 76-77: natural wood production forests. I don't understand. Define the term natural in this context? Isn't that naturally regenerated or seeded/planted by original tree species (mixture) of the vegetation type in question. Same chapter: native forest. Don't understand. Semi-natural?

A short chapter describing the differences in the understory vegetation (the forest floor: bottom and field layers), usually very different in plantations than in naturally regenerated forests. I guess can generalize that plantations usually host grass-dominated (often invasive species) understory, compared to naturally regenerated forest stands, which leads to different fire risk, as ignition probably is higher with these flash fuels, as grasses in the plantations. So how I see this, it's not the tree species, but understory vegetation merely that causes forest fires, as ignition most often happens in the forest floor. This mainly causes the differences in fire statistics between plantations and naturally developed forests. Naturally changes in the understory vegetation in both naturally regenerated forests and plantations vary according to biomes and vegetation types that will lead differences in fire risk and fire statistics. Invasive species cause remarkable influences to fire risk, number of fires, and area burnt, as can be seen e.g. in Indonesia.

Eucalyptus and Pinus genera included in the same discussion when estimating fire risks and damages could also be opened a bit? They (depending on the species of course) do have different litter accumulation, causing different flammability and fire risk, especially if looking for stand-replacing (would use instead stand-destroying) fires in different countries using different species in plantations, which may partly cause the differences?

Including few lines to describe these differences would strengthen the ms.

Few detailed comments:

line 189-190: probably not forest type? In forestry it has a different meaning. You probably mean "forest production type" (as defined in this ms?). Please clarify the definitions.

lines 262-264: ... plantation expansion should be concentrated in areas, where fire risk is low, Northern countries...

I strongly disagree with your conclusion for two reasons. Productivity in Northern countries is lower, and it would lead to destroying local nature by using non-original tree species, being followed by other invasive species, most likely. I strongly advice you to remove this sentence or re-formulate it.

Line 340: research misspelled.

This is a nice attempt to look plantation forests and forest fires together. As the paper is global (general) and the used methodology and analyses sound, I don't have much to improve the paper, and suggest it to be accepted with minor changes.

Version 1:

Reviewer comments:

Reviewer #1

(Remarks to the Author)

I am very happy with the authors response to my review comments. They have addressed them comprehensively.

REVIEWER COMMENTS

Reviewer #1 (Remarks to the Author):

This paper addresses an increasingly common question about fire proneness of various types of forests and it's very good to see an attempt to compare natural forest systems with plantations. The use of the Lesiv approach to access data on the different management types of forests gives a good opportunity to address this question globally and the matching approach allows proper comparisons under similar conditions.

The paper is important as it is the first attempt at comprehensively answering the question as to whether plantations have a higher fire incidence than natural production forests under the same conditions– but it requires some caveats.

We thank the Reviewer for their extremely useful and constructive comments which we feel have helped to improve the manuscript significantly.

I query the data for Belarus and Ukraine and their reported proportion of plantations in Figure 3. Looking at FRA data for 2015 Belarus has only reported 6040 hectares of plantation and a total forest area of 8.633 m hectares – or 0.1% plantations, Ukraine has 368,000 ha of plantations with 9.65 m hectares of total forest or 3.8%. Some variation is to be expected as resource data sources are different (FRA and Lesiv) – but this is very large.

New Zealand is also mis represented – plantation timber production makes up nearly 99% of national production – natural forests in public ownership are managed purely for conservation purposes and only a very small area of privately owned natural forest is managed for timber. This suggests misclassification in the Lesiv paper.

We thank the Reviewer for the important point raised regarding discrepancies between the Lesiv et al and FAO data. We believe the discrepancies between the numbers reported by Lesiv *et al.* and the FAO FRA for countries such as Belarus and Ukraine are largely due to different definitions of forest types and in particular the difference in definitions between 'planted forests' and 'plantation forest'.

The FAO define 'planted forest' as "a forest that at maturity is composed predominantly of trees established through planting and/or deliberate seeding", whereas Lesiv *et al.* define a 'planted forest' as "managed and there are signs that the forest has been planted in the 100 m pixel. Rotation time is relatively long (>15 years)". The FAO's

definition suggests planted forest would not be primarily used for timber production, whereas Lesiv's et al's definition suggests the opposite. In this study, we take Lesiv et al's definition, and use 'planted forest' to represent temperate plantations (since Lesiv et al map no 'plantation forests' in the temperate regions). When considering total area of 'planted forest', in 2020 Belarus and Ukraine had 1.9 and 4.9 Mha, respectively. This is not too dissimilar to Lesiv *et al.*'s estimate of 3.9 and 5.9, Mha.

To make this clearer, we have included a definitions table in the revised Methods section that sets out clear definitions for 'temperate' and 'tropical' plantations used in this study, and have also included a Supplementary Table that outlines the areas of natural production forest and plantation in this study, and compares them to reported areas of equivalent forest type in the FAO FRA.

Regarding New Zealand, we appreciate the Reviewer's point that the vast majority of timber comes from plantations. We have now updated the main text to reflect this point:

L210-213: *"Other countries less vital to global timber production have similar risk profiles given their reliance on plantations, including Ukraine, Belarus, and Portugal, whilst restrictions on natural forest harvesting means that >99% of timber in New Zealand is now sourced from plantations (NZFOA, 2023)."*

It would be helpful and give readers context if there was a summary table of area of naturally regenerating production forest, tropical plantations and temperate plantation (the three management types used) for the countries in a supplementary table. I suggest review of Figure 3 and working out how best to make the important point about the observed increasing risk with increasing dependence on plantation sourced timber.

We think it is a good idea to include a summary table to help the reader understand the national-level split and total areas of the different timber production types. We have now included this Table in the supplementary material, and also included a comparison between our area results based on the Lesiv *et al.* data and those reported in the 2015 FAO FRA. We have edited Figure 3 to more clearly mark out countries such as New Zealand, China, Vietnam, and Australia who either already source all of their timber from plantations, or will do so in the near future. We think it makes sense to keep the plantation proportion data using the Lesiv *et al.* dataset as the source for Figure 3 given that our analysis has focused on this dataset. Nevertheless, we have now also included another version of the figure in the supplementary material that instead displays the proportion plantation according to the 2020 FAO FRA data for comparison.

Supplementary Table 1. Comparison between total area of each timber production type from the Lesiv *et al.* dataset and the 2015 FAO FRA. Percentages represent the percentage of a country's timber-producing forest that production type represents. 'Other naturally regenerated forest' best represents the natural production forest assessed in this study, but area extents are likely greater in the FAO FRA due to this category also including recovering forests on agricultural land. *China is mapped by Lesiv *et al.* as having a combination of temperate and tropical plantations, so the value in this table is the combined extent of both plantation types.

Country	Natural production forest area (Mha)	Plantation area (Mha)	Plantation type	FAO FRA 2015 'Other naturally regenerated forest' area	FAO FRA 2015 'Planted forest'
United States	239,931 (88%)	33,097 (12%)	Temperate	208,431 (89%)	26,364 (11%)
China	106,499 (55%)	88,111* (45%)	Temperate	117,707 (60%)	78,982 (40%)
Brazil	177,251 (94%)	10,432 (6%)	Tropical	283,111 (97%)	7,736 (3%)
Russia	246,934 (90%)	26,528 (10%)	Temperate	522,372 (96%)	19,841 (4%)
Indonesia	34,418 (94%)	2,158 (6%)	Tropical	40,040 (89%)	4,946 (11%)
Vietnam	9,358 (63%)	5,451 (37%)	Tropical	11,027 (75%)	3,663 (25%)
France	14,990 (73%)	5,527 (27%)	Temperate	15,022 (88%)	1,967 (12%)
New Zealand	3,299 (69%)	1,494 (31%)	Temperate	5,905 (74%)	2,087 (26%)
Australia	53,809 (95%)	2,778 (5%)	Tropical	117,695 (98%)	2,017 (2%)
South Africa	16,169 (94%)	992 (6%)	Tropical	6,531 (79%)	1,763 (21%)
Belarus	6,003 (60%)	3,945 (40%)	Temperate	6,323 (77%)	1,910 (23%)
Spain	11,814 (80%)	2,903 (20%)	Temperate	15,509 (84%)	2,909 (16%)
Argentina	40,143 (96%)	1,791 (4%)	Tropical	24,172 (95%)	1,202 (5%)
Uruguay	1,560 (58%)	1,150 (42%)	Tropical	470 (31%)	1,062 (69%)
Ukraine	6,383 (52%)	5,904 (48%)	Temperate	4,738 (49%)	4,860 (51%)
Portugal	1,759 (68%)	837 (32%)	Temperate	2,267 (72%)	891 (28%)
Italy	10,376 (96%)	449 (4%)	Temperate	8,565 (93%)	639 (7%)

Supplementary Figure 2. National-level plantation wildfire risk effect size and percentage of timber-producing forest under plantation systems. Plantation stand-replacing wildfire risk represents the relative mean percentage change in wildfire probability in plantations compared to natural production forests under the same wildfire conditions. Points in orange to red represent increasingly high plantation wildfire effect (i.e. plantations are increasingly more likely to suffer stand-replacing wildfire than natural production forest), points in darker blue represent increasingly low plantation burn effect (i.e. plantations are increasingly less likely to suffer stand-replacing wildfire than natural production forest), and points in grey demonstrate countries with no significant plantation burn effect. Percentage of timber-producing forest under plantation systems is from the 2020 FAO Forest Resources Assessment³. Points are sized by their share of global timber production between 2015-2022 according to the FAO²¹. Countries with * are those that have banned native forest logging either at the national (China, Vietnam, New Zealand) or state level (Australia) and source the majority of their timber from plantations.

The paper should include some comment in the discussion about uncertainty in the findings – the Levis paper notes it is ‘the first reference data set and a prototype of a globally consistent forest management map’ and ‘with forest management class accuracies ranging from 58% to 80%’ – and this uncertainty will carry through to the analysis in this paper. See my comments on differences between this data and FRA data above.

We agree that more should be made about the inherent uncertainties in the Lesiv et al. map, which will follow through into our own analysis. Alongside additional discussion of FAO issues, we now address this in the text:

L218-233: *“The map of forest management¹⁹ allows for global-scale comparisons of fire risk in natural production forests and plantations, but mapping timber harvests globally at such fine resolution is difficult, and these data are not without uncertainty. Whilst the overall accuracy of the map is 83%, the user’s accuracy for the forest management classes used in this study range from 58-71%, with the extent of temperate plantations often underestimated and confused with natural production forests. To counter this, we included only mapped pixels with a classification confidence $\geq 70\%$, which represented a compromise between high-confidence classifications and ensuring enough data points to allow for rigorous statistical matching and modelling. Mapping of different timber production systems is also difficult due to ambiguous definitions of production types within the FAO framework, particularly in the case of ‘planted forests’ and ‘plantation forests’ where the specific differences between the two, and whether both typically produce timber, is not always clear. The FAO definition of forest consisting of only 10% tree cover also raises difficulties, since this definition will include many savannah areas (Parr et al. 2014) with limited timber value. Moving towards improved and more specific definitions of forest management, alongside globally consistent annual satellite-based mapping of timber harvests, would allow for dramatic improvements in our understanding of timber harvest dynamics and future threats to global timber production.”*

I found Figure 1 to be too small to properly view the images and suggest it be split into two.

We appreciate the Reviewer’s concerns about the size of the panels in Figure 1. We agree they are small in word document format but suggest that in the online and printed versions of the final manuscript they will be large enough for viewing. The current format (2 by 2 grid) makes it much easier to make comparisons between the global distribution of natural production forests compared to plantations, and also make comparisons between burned area patterns. We have attached an alternative, taller version of the figure to this document which contains larger panels at the expense of easy comparisons between the four panels. With the Editor’s and Reviewer’s permission we would prefer to keep the figure as it is currently but will defer that decision.

Be consistent – natural or native production forests

Thank you for pointing out the inconsistencies in terms. We have now modified the manuscript to ensure that we consistently refer to natural production forests when discussing the production type used in our analysis, and define these in the Methods. Some references to native forests remain, since we occasionally discuss them separately from the wood production context.

Line 85. Potentially modify this to say plantations are low diversity often

containing one or two species initially planted in densely packed rows as species source may not be all that relevant in the description it's not so much native versus exotic as species fire susceptibility.

We have edited the line as suggested, it now reads:

L86-87: *"Plantations are low diversity, often containing one or two species initially planted in densely packed rows"*

Reviewer #2 (Remarks to the Author):

Bousfield et al. examined wildfire risks across timber production systems in the major timber production counties in the world. The forests were divided into the natural production forest and plantation forest. GAMMs was fitted and implemented to estimate the burn probability. Forest management data was from Lesiv et al., which contains data on forest management type across the world forest at 100 m resolution. The method is sound. My major concern is how this paper is different to their previous work published in 2023.

Bousfield, Christopher G., David B. Lindenmayer, and David P. Edwards. "Substantial and increasing global losses of timber-producing forest due to wildfires." *Nature Geoscience* 16, no. 12 (2023): 1145-1150.

While the broad focus of this manuscript is on wildfire, it is tackling a totally different question to Bousfield et al. (2023). First, the central question of this paper is focused on quantifying differences in burn risk between different timber production systems (i.e. natural versus managed production forest), whereas Bousfield et al. documented the increasing trends in annual burned area across the entire global extent of timber-producing forests regardless of timber production method.

Second, this paper uses novel statistical matching and modelling approaches to control for other fire-influencing factors and isolate the impact of timber production type on burn risk. By contrast, Bousfield et al. used an entirely different methodological approach relying upon descriptive spatial methods that map the extent of wildfire and simple regression to detect temporal trends in wildfire extent within timber-producing forests. The strong focus of this manuscript is thus on identifying the drivers of burn risk patterns between production types, and not simply describing general burn patterns across all timber production forest.

In Figure 1, it is not clear what the blue color in C and D denote.

We refer to the blue colour in the figure legend, and state that:

L128-129: *"areas mapped in blue have no or limited burned area (<0.1%) in timber-producing forest"*

Reviewer #3 (Remarks to the Author):

Article "Global risk of wildfire across timber production systems" deals with a very interesting topic. The questions raised in the article are indeed worth of discussion worldwide. The article fits well in the scope of the journal and is indeed worth publishing. I feel this will increase people's understanding of the role of fires in forestry and in relation with global wood markets. The paper is generally well-written, used analyses sound, and the given results rely on the literature cited well.

We thank the Reviewer for their helpful comments and suggestions that have significantly improved our manuscript.

However, I include number of suggestions that might improve the paper.

I suggest to include few lines & include critical discussion about FAO:s forest classification (that do have many inaccuracies) that cause some possibilities also to have errors when interpreting global forest data. FAO:s forest classification is based on too loose definitions.

Terminology used in this ms is somewhat confusing, and for exact definitions, you need to look for them in the methods-section. Ms could be improved by being more exact with the terminology. A simple table with terminology & definitions used might help the reader. Or at least short definition in the main text. Suggest using the defined terminology throughout the paper. An example, lines 76-77: natural wood production forests. I don't understand. Define the term natural in this context? Isn't that naturally regenerated or seeded/planted by original tree species (mixture) of the vegetation type in question. Same chapter: native forest. Don't understand. Semi-natural?

We appreciate the confusion regarding terminology and definition of different production forest types. We have adopted the Reviewer's suggestion and included a definitions table in the Methods section, which includes the term we use in the study, the corresponding map class in Lesiv *et al.*, and the original definition from Lesiv *et al.* when making their forest management map. We have signposted this Table in the main text

when first referring to the forest management types, and hope this makes the terminology of the manuscript clearer.

Regarding the use of natural wood production forests, we have edited this line to make clearer that we are referring to naturally occurring native forests that are harvested and then regrow naturally, as opposed to intensively managed timber plantations.

The text now reads:

L76-78: *"There are two primary methods of producing timber: through the harvesting of naturally occurring native forests that produce wood and regrow naturally after logging, or through intensively managed timber plantations composed of one or two species of even ages with regular spacing"*

We also now include additional discussion about the issues with current FAO terminology:

L226-234: *"Mapping of different timber production systems is also difficult due to ambiguous definition of production types within the FAO framework, particularly in the case of 'planted forests' and 'plantation forests' where the specific differences between the two, and whether both typically produce timber, is not always clear. The FAO definition of forest consisting of only 10% tree cover also raises difficulties, since this definition will include many savannah areas (Parr et al. 2014) with limited timber value. Moving towards improved and more specific definitions of forest management, alongside globally consistent annual satellite-based mapping of timber harvests, would allow for dramatic improvements in our understanding of timber harvest dynamics and future threats to global timber production."*

A short chapter describing the differences in the understory vegetation (the forest floor: bottom and field layers), usually very different in plantations than in naturally regenerated forests. I guess can generalize that plantations usually host grass-dominated (often invasive species) understory, compared to naturally regenerated forest stands, which leads to different fire risk, as ignition probably is higher with these flash fuels, as grasses in the plantations. So how I see this, it's not the tree species, but understory vegetation merely that causes forest fires, as ignition most often happens in the forest floor. This mainly causes the differences in fire statistics between plantations and naturally developed forests. Naturally changes in the understory vegetation in both naturally regenerated forests and plantations vary according to biomes and vegetation types that will lead differences in fire risk and fire statistics. Invasive species cause remarkable influences to fire risk, number of fires, and area burnt, as can be seen e.g. in Indonesia.

Eucalyptus and Pinus genera included in the same discussion when estimating fire risks and damages could also be opened a bit? They (depending on the species of course) do have different litter accumulation, causing different flammability and fire risk, especially if looking for stand-replacing (would use instead stand-destroying) fires in different countries using different species in plantations, which may partly cause the differences?

Including few lines to describe these differences would strengthen the ms.

Thank you for your suggestions regarding mechanisms for increased fire risk in plantations. We have included additional discussion to develop the point about plantation species influencing burn risk. We discuss studies that have found differing burn rates between pine and eucalyptus plantations, and indicate that understanding the influence of plantation species on burn risk is an important research question going forward. We also included an additional discussion of plantation understories and the prevalence of exotic species in plantations which can increase fire risk, although we found the literature lacking detail in this regard. The text now reads:

L267-275: *"In addition, low-diversity understories in exotic timber plantations (Wang et al. 2022; Calviño-Cancela et al. 2012) can harbour invasive species (Paritsis and Aizen, 2008), which increase fire risk (Fusco et al. 2019). The dominant tree species used also plays a role in fire proneness. In 2017, record-breaking wildfires in Chile burned greater proportions of Pinus plantations than Eucalyptus (Bowman et al. 2018), whilst in the record 2019/20 Australian wildfire season, conifer plantations suffered higher incidences of fire-induced severe canopy damage compared to Eucalyptus plantations, potentially due to the higher live fuel load and density in Pinus tree crowns (Bowman et al. 2021). Understanding potential variance in fire proneness between different plantation species is therefore an important next step in minimising fire risk across global plantation systems."*

Regarding the use of 'stand-destroying' fires instead of 'stand-replacing', we would prefer to continue referring to them as 'stand-replacing' since this is the original terminology used in the Tyukavina *et al* paper that produced the fire dataset, but we are happy to defer this decision to the Editor.

Few detailed comments:

line 189-190: probably not forest type? In forestry it has a different meaning. You probably mean "forest production type" (as defined in this ms?). Please clarify the definitions.

Thank you for pointing this mistake out. We have now corrected it to read:

L192-193: *“Of the remaining tropical countries, Brazil, Argentina, and Vietnam demonstrated no effect of forest production type on the likelihood of stand-replacing wildfires”*

lines 262-264: ... plantation expansion should be concentrated in areas, where fire risk is low, Northern countries...

I strongly disagree with your conclusion for two reasons. Productivity in Northern countries is lower, and it would lead to destroying local nature by using non-original tree species, being followed by other invasive species, most likely. I strongly advice you to remove this sentence or re-formulate it.

We understand the Reviewers concern about lack of productivity in the northern countries and the potential damage to natural ecosystems of plantation expansion. We have edited these sentences so they now read:

L289-292: *“At the global scale, plantation expansion should be concentrated in areas where productivity is high, and absolute fire risk is relatively low⁵, provided plantation expansion does not replace natural ecosystems¹², and could instead occur on low-intensity or abandoned agricultural land.”*

Line 340: research misspelled.

Thank you, we have now corrected this typo.

This is a nice attempt to look plantation forests and forest fires together. As the paper is global (general) and the used methodology and analyses sound, I don't have much to improve the paper, and suggest it to be accepted with minor changes.

We thank you for your useful and constructive comments.